# The Interplay between *Candida albicans,* Vaginal Mucosa, Host Immunity and Resident Microbiota in Health and Disease: An Overview and Future Perspectives

**DOI:** 10.3390/microorganisms11051211

**Published:** 2023-05-05

**Authors:** Roberta Gaziano, Samuele Sabbatini, Claudia Monari

**Affiliations:** 1Department of Experimental Medicine, University of Rome Tor Vergata, 00133 Rome, Italy; 2Department of Medicine and Surgery, Medical Microbiology Section, University of Perugia, 06132 Perugia, Italy; samuele.sabbatini@unipg.it (S.S.);

**Keywords:** vulvovaginal candidiasis, *C. albicans*, vaginal microbiota, host’s immune response, vaginal inflammation, *Candida* virulence factors, probiotics, vaginal microbiota transplantation

## Abstract

Vulvovaginal candidiasis (VVC), which is primarily caused by *Candida albicans*, is an infection that affects up to 75% of all reproductive-age women worldwide. Recurrent VVC (RVVC) is defined as >3 episodes per year and affects nearly 8% of women globally. At mucosal sites of the vagina, a delicate and complex balance exists between *Candida* spp., host immunity and local microbial communities. In fact, both immune response and microbiota composition play a central role in counteracting overgrowth of the fungus and maintaining homeostasis in the host. If this balance is perturbed, the conditions may favor *C. albicans* overgrowth and the yeast-to-hyphal transition, predisposing the host to VVC. To date, the factors that affect the equilibrium between *Candida* spp. and the host and drive the transition from *C. albicans* commensalism to pathogenicity are not yet fully understood. Understanding the host- and fungus-related factors that drive VVC pathogenesis is of paramount importance for the development of adequate therapeutic interventions to combat this common genital infection. This review focuses on the latest advances in the pathogenic mechanisms implicated in the onset of VVC and also discusses novel potential strategies, with a special focus on the use of probiotics and vaginal microbiota transplantation in the treatment and/or prevention of recurrent VVC.

## 1. Introduction

*Candida albicans* is the most common fungal pathogen associated with opportunistic infections in humans [1,2]. As a member of the human mycobiota, *C. albicans* generally colonizes the mucosal surfaces of mosthealthy individuals. Under normal physiological conditions, mucosal sites are characterized by a delicate and complex balance between *Candida*, host immunity and local microbiota. In fact, both the immune response and microbiota composition play a central role in counteracting overgrowth of the fungus and maintaining host homeostasis [3]. Dysfunction of the host immune defense mechanisms or alteration in microbial composition may promote fungal virulence and, thus, increase the risk of *Candida* infections. A wide spectrum of diseases is caused by *Candida* spp., ranging from superficial muco-cutaneous to systemic life-threatening candidiasis, which is especially prevalent in severely immunocompromised patients. Among the known superficial mycotic infections, vulvovaginal candidiasis is the second most common cause of vaginitis worldwide. It has been estimated that VVC affects 75% of all women at least once in their lifetime, and it is predominant in women of reproductive age [4]. When >3 episodes of VVC occur per year, the condition is diagnosed as RVVC, which affects nearly 8% of women globally [5].

Various well-known risk factors predispose individuals to VVC, including wide-spectrum antibiotic therapy, certain sexual behaviors, oral hormonal contraceptives, pregnancy and diabetes mellitus. However, the risk factors for RVVC are currently not fully defined, although several studies have highlighted a genetic component in the susceptibility to RVVC infections. Specifically, genetic variations in Toll-like receptor (TLR)-2, which belongs to the family of pattern recognition receptors (PRRs) and is expressed on the cellular surface of innate immune cells, seem to be associated with RVVC [6]. Moreover, a variable-number tandem repeat in the NLRP3 inflammasome, which leads to a hyper-inflammatory neutrophil response to *Candida* colonization in the vaginal mucosa, is more likely to play a critical role in the onset of RVVC than defective host immune response [7,8]. Indeed, contrary to other forms of candidiasis, such as oral or systemic *Candida* infections, VVC and RVVC also occur in immunocompetent and otherwise healthy women. This supports the notion of a genetic predisposition for susceptibility to RVVC. Further, virulence factors linked to genetic and phenotypic intra-species variations in *C. albicans* might also play a significant role in the development of vaginal candidiasis [9].

This review provides an overview of the current knowledge regarding the pathogenic mechanisms underlying VVC/RVVC, with particular emphasis on the key role played by host innate immunity, vaginal dysbiosis and *Candida*-related virulence factors. We also discuss novel potential therapeutic strategies based on the use of probiotics and vaginal microbiota transplantation (VMT) as alternative or adjunctive interventions to conventional antifungal drugs for the prevention and treatment of VVC/RVVC. Even though vaginal candidiasis is not a life-threatening disease, its high incidence rate worldwide, along with its strong negative impact on women’s quality of life, requires further understanding of the complex interaction among host immunity, vaginal microbiota and *C. albicans*, for the design of novel potential therapeutic approaches.

## 2. Role of Vaginal Epithelial Cells in the Pathogenesis of VVC

### 2.1. Vaginal Epithelial Cells: More Than a Mere Barrier

The vaginal mucosa is not merely a physical barrier, but it actively participates in host immune defense against the invasion of underlying tissue by potential harmful pathogens, such as *Candida* spp. To this end, vaginal epithelial cells (VECs) express on their surface a class of receptors called pattern recognition receptors (PRRs), via which VECs are able to recognize and bind specific pathogen-associated molecular patterns (PAMPs) secreted by or presented on the surface of human pathogens, including *C. albicans* [10,11,12]. Particularly, VECs sense microbial PAMPs using several PRRs belonging to the TLRs, C-type lectin receptor (CLR), nucleotide oligomerization domain-like receptor (NLR) and RigI-helicase receptors families. Among these, TLR2, TLR4, Dectin-1 and the non-classical PRR ephrin type-A receptor 2 are involved in the recognition of *C. albicans* [3].

Epithelial cells also release antimicrobial peptides in response to pathogens: for example, iron-binding compounds, such as neutrophil gelatinase-associated lipocalin, lactoferrin and calprotectin, display broad-spectrum antimicrobial activity, as well as antifungal properties, by inhibiting the proliferation of microorganisms that require iron for their growth [13]. Some other antimicrobial factors released by VECs include mannose-binding lectin (MBL), beta defensins and cathelicidins. Indeed, women with MBL deficiency, which is caused by genetic polymorphisms, are known to be more susceptible to RVVC [14].

Exfoliated and lysed epithelial cells release glycogen into the vaginal lumen. The glycogen is catabolized by α-amylase and leads to the production of small carbohydrates, which in turn, favor the proliferation of lactobacilli that are essential for vaginal health [15]. Finally, the rapid regeneration of squamous epithelial cells in the mucosal lining has a protective effect, as this counteracts the adhesion and invasion of *C. albicans*, which needs to adhere to the mucosal lining for establishing vaginal infections. Thus, besides their function as a physical barrier, VECs are also considered as initial sensors of danger and executors of appropriate defense responses against pathogens, such as *C. albicans*, as well as a link between the innate and adaptive immune systems [16].

### 2.2. Response of VECs to C. albicans

The adhesion of *C. albicans* to the vaginal mucosal surface is essential for its persistence in the host either as a commensal or pathogenic microorganism. This is a complex, dynamic, multifactorial process involving an intimate association between components of the fungal cell wall and epithelial surface proteins. VECs are continuously exposed to *C. albicans*, as this fungus is the main component of the healthy vaginal mycobiota. Interestingly, it has been demonstrated that VECs are able to discriminate between commensal and pathogenic forms of *C. albicans*. Moreover, the host innate immune response is dependent on both *Candida* morphotype and fungal load. Indeed, in healthy women, VECs are able to tolerate *C. albicans* yeast and low filamentous fungal burden. However, in the presence of a massive load of hyphae, VECs become susceptible to the fungus and trigger an intense inflammatory response [10,17,18,19,20,21]. The initial interaction between the fungus and the vaginal epithelium is mediated by passive forces, i.e., van der Waals forces and hydrophobic interactions. Subsequently, the adhesion is strengthened and stabilized by the binding of the fungal adhesins with specific surface epithelial cell receptors and host extracellular matrix components [19,20,22]. To facilitate its adhesion to the receptors, *C. albicans* possesses a large variety of adhesins, including the agglutinin-like sequence (Als) protein family, which are well characterized. Eight members of this family (Als1p–Als7p and Als9p) are glycosylphosphatidylinositol-linked to 1,6-glucans present on the fungal cell wall. The *ALS1-5* and *ALS9* genes were found to be consistently expressed in an experimental model of oral candidiasis [23]; in contrast, the *ALS1*, *ALS2*, *ALS3* and *ALS9* genes were frequently detected in clinical samples of vaginal fluid [24]. These findings imply that different proteins in the Als family play specific roles in different parts of the mucosal surface.

Once *Candida* has adhered to the vaginal mucosa, the dimorphic transition from the yeast to the hyphal form results in the expression of additional adhesins associated with the hyphal morphotypes, namely, Als3p, Hwp1p (hyphal wall protein 1) and Ssa1p. These proteins reinforce the adhesion of the fungus to epithelial cells by interacting with E-cadherin, a key component of intercellular junctions [25], and with a heterodimeric receptor complex composed of epidermal growth factor receptor (EGFR) and Her2 (EGFR/Her2) [25,26]. Targeting the virulence factors has become a new research direction to combat *Candida* infections. In particular, in recent years, Als3p has been considered as a potential target of vaccines against *C. albicans* infection, and an NDV-3 vaccine targeting the Als3p N-terminal has recently entered the clinical trial phase [27].

Subsequent to adhesion, *Candida* hyphae can penetrate epithelial cells through two distinct mechanisms: endocytosis and active penetration. The former is induced by the interaction of the *C. albicans* proteins Ssa1p and Als3p with cadherins and EGFR/Her2 [25,26,28,29], while the latter occurs when the growing hyphal tip pushes the epithelial cell membrane and leads to cell damage through the release of various virulence factors, including the cytolytic peptide toxin candidalysin, secretory aspartyl proteinases (SAPs), lipases and phospholipases. Recognition of the filamentous forms of *Candida* by VECs leads to sustained activation of all three mitogen-activated protein kinase (MAPK) pathways (p38, JNK and ERK1/2), which in turn, leads to c-fos activation via p38 and release of antimicrobial peptides, such as alarmins and pro-inflammatory cytokines (GM-CSF, G-CSF, IL-6, IL-1α, IL-1β, IL-36γ and CCL20) that are essential for the recruitment of innate immune cells, mainly neutrophils and macrophages [18,20,30,31,32,33,34]. Several recent investigations showed that fungal components, i.e., candidalysin [35,36,37] and SAPs, as well as the intracellular damage-responsive molecular complex, called the NLRP3 inflammasome, of VECs [38,39,40,41] are all critical actors in VVC immunopathology. Indeed, both candidalysin and SAPs have been found to trigger strong and persistent activation of the NLRP3-caspase-1 pathway to lead to the production of pro-inflammatory cytokines, particularly IL-1β (the major inflammasome effector), and recruitment of neutrophils, which (however) appear to be unable to eradicate the fungal infection and, actually, further contribute to the perpetuation of the chronic inflammatory status [20,42]. This is discussed in more detail in the next section.

It has been demonstrated that the type I IFN signaling, induced by vaginal epithelial cells in response to *Candida* spp. infections, increases epithelial resistance to infection and dampens pro-inflammatory responses [43]. An interesting work conducted by Sala et al. analyzed a set of VVC and colonizing *C. albicans* vaginal isolates to identify in vitro phenotypes associated with one group or the other. Interestingly, the isolates did not differ in terms of genetic profile or behavior (growth rate, and filamentation) in culture media but they exhibited different behaviors during their interactions with VECs. Specifically, *C. albicans* isolates from VVC induced more fungal shedding from epithelial cells and increased epithelial cell detachment than those from healthy women. Of particular interest, VVC-associated isolates also failed to elicit type I IFN. This study highlights that VVC isolates may have intrinsically greater pathogenic potential by means of their ability to elicit different epithelial responses, including the type I IFN pathway [44]. A schematic representation of the principal effects resulting from the *C. albicans* hyphae-VECs interaction is shown in Figure 1.

## 3. Host Immune Response in VVC Pathogenicity

### 3.1. Vaginal Host Immune Response to C. albicans

Whilst the adaptive immune response represents a crucial host defense mechanism against systemic and oral candidiasis, innate immunity plays a major role in immune protection against VVC. The pro-inflammatory factors released by VECs in response to *C. albicans* induce the recruitment of professional phagocytes, such as neutrophils, macrophages and dendritic cells (DCs), to the site of infection. Phagocytic cells are able to recognize pathogens through a wide spectrum of PRRs, including TLRs and several CLRs, such as Dectin-1, Dectin-2 and the mannose receptor, which are considered as major players in the recognition of *C. albicans* by VECs [45]. Once stimulated by *Candida* PAMPs, PRRs activate intracellular signaling pathways, orchestrating the inflammatory immune response against the fungus [46,47]. In fact, as for VECs, TLR-2 and TLR-4 recognize fungal mannoproteins, while the cytosolic nucleotide oligomerization domain-2, along with TLR-9, senses fungal chitin as a PAMP [48]. Dectin-1 is the major β-glucan receptor in myeloid cells, and it plays an important role in host defense against fungi. The recognition of Dectin-1 triggers a plethora of downstream signaling pathways, with complex interrelationships. Indeed, Dectin-1 has been implicated in mediating/regulating several immune responses to *C. albicans*, including phagocytosis, inflammasome activation, cytokine/chemokine production, respiratory burst, inflammatory cell recruitment, neutrophil extracellular trap (NET) and T cell responses [49,50,51,52]. Dectin-2 recognizes α-mannan of fungal pathogens and is involved both in cytokine production, through Syk-CARD9-NF-κB signaling [53,54], and in NET release in response to un-opsonized *C. albicans*, through an NADPH oxidase-independent pathway [55]. In addition to these proteins, CARD9 is an adaptor protein implicated in the transduction of signals from a variety of PRRs, including TLRs and CLRs. A homozygous mutation of this protein is associated with predisposition to both mucosal and systemic candidiasis [56]. Moreover, genetic polymorphisms in PRRs, especially in the TLR2 gene, have been associated with increased susceptibility to VVC caused by defective production of IFN-γ and IL-17, which are crucial in the host defense against fungal infections [6].

Among phagocytic cells, neutrophils are considered the most effective cell type in controlling *Candida* infection [57,58], as they can kill both the yeast and hyphal forms through intracellular and extracellular killing mechanisms, respectively. The intracellular mechanism is mediated by phagocytosis, hydrolytic enzymes, antimicrobial peptides and reactive oxygen species (ROS), while the extracellular mechanism involves the release of NET, autophagy and ROS activation [59,60,61,62,63]. In fact, we demonstrated [63] that, when neutrophils were incubated with *C. albicans*, the hyphal cells were able to induce the production of large quantities of NET as early as 15 min, and this effect persisted for a 4 h period. Our findings indicated that autophagy is involved in both rapid and late NET release, whereas ROS production occurs only after 4 h of neutrophil incubation with *Candida*. However, Zambrano et al. [64] recently demonstrated that while *C. albicans* triggers NET formation in the vaginal mucosa, it is very likely that the fungus may elude the effects of NET through the release of DNase and mucosal biofilm formation. In addition, since neutrophils are killed during NETosis, the subsequent release of their nuclear and cytoplasmic contents may help promote vaginal inflammation and, thereby, lead to tissue damage.

Overall, despite the massive infiltration of neutrophils into the vaginal lumen, they not only are unable to clear *C. albicans* cells, but also worsen the fungus-mediated damage and promote a hyper-inflammatory loop that confines the vaginal mucosa to a chronic inflammatory state [3,10,20,21,33,65,66,67,68]. However, despite extensive research efforts, the reasons for the impaired neutrophil effector activity remain a mystery and are currently the subject of considerable research on the pathogenesis of VVC, which is presently an enigmatic infection. Recent findings have shed light on the mechanisms underlying the neutrophil dysfunctionality, and one of the mechanisms is the presence of specific factors that can inhibit neutrophil effector functions in the vaginal environment during VVC. In particular, Yano et al. [42,69] demonstrated that the proteoglycan heparan sulphate, which is present at high levels in the estrogen-responsive state, is a competitive inhibitor of the interaction between neutrophils and *Candida*. In addition, Ardizzoni et al. [10,70] showed that perinuclear anti-neutrophil cytoplasmic antibodies, which are present at high levels in women with VVC, in comparison to healthy women, drastically reduce the ability of neutrophils to eliminate the fungus.

Macrophages contribute to fungal clearance through uptake of fungal cells. They display greater phagocytic capacity than neutrophils, but their uptake rate and intracellular killing ability is lower than that of neutrophils [71,72]. Indeed, the hyphal form of *C. albicans* is relatively resistant to phagocytosis and is able to escape macrophages via several strategies, such as permeabilizing macrophage membranes via candidalysin and engaging two host cell death pathways, namely, Gasdermin D-mediated pyroptosis and ETosis [73,74]. These processes not only facilitate the immune evasion of *C. albicans*, but also trigger inflammation. In fact, the suppression of hyphal escape reduces inflammatory activation [73]. These results imply that macrophages may also play a role in the immunopathogenesis of VVC. Accordingly, future studies on the pathogenesis of VVC should also focus on fungal escape from macrophages.

While the successful clearance of *C. albicans* during VVC strongly depends on the phagocytosis of the fungal pathogen by innate immune cells, the protective role of adaptive immunity in vaginal candidiasis remains somewhat unclear. In fact, susceptibility to oral, chronic mucocutaneous, and gastrointestinal candidiasis has been clearly associated with deficiencies in cell-mediated immune response. However, numerous studies carried out in women affected by VVC revealed no significant protection provided by local or systemic adaptive immune response [75,76,77] and, in particular, the role of Th17 cells is currently controversial. Indeed, several studies in experimental animal models have shown [78,79,80] that these cells have a protective role against vaginal candidiasis. Moreover, evidence from human studies also supports the role of the Th17 pathway in the immune response to *Candida* infections. In fact, an increased risk for fungal infections, including cutaneous and genitourinary candidiasis, was observed in patients with psoriasis treated with IL-17 inhibitors [81]. On the other hand, recent evidence suggests that the Th17/IL-17 pathway plays a non-essential role in the immunopathogenesis of VVC. In accordance with these findings, Yano et al. [82] used genetically deficient mouse strains and showed that IL-17 did not play a role in neutrophil infiltration during VVC, and Peters et al. [83] also demonstrated that the Th17/IL-17/IL-22 immune axis is dispensable for antifungal defense at the murine vaginal interface.

### 3.2. NLRP3 Inflammasome Dysregulation as a Key Factor in the Immunopathogenesis of VVC

Accumulating evidence has highlighted the critical role of the NLRP3 inflammasome in the molecular mechanism behind the immunopathogenesis of VVC [40]. The activation of the inflammasome, both in the vaginal epithelium as well as in innate immune cells [84], is involved in host defense mechanisms against *Candida* infection, as the activated inflammasome mounts an effective immune response and, thus, limits fungal burden and inflammation. Various cell wall components in fungi, through their interaction with specific PRRs expressed on phagocytes, provide a priming signal, leading to the synthesis of pro-IL-1β and NLRP3 inflammasome components. For instance, the binding of β-glucan and zymosan to the CLR Dectin-1 and TLR2 has been proven to prime macrophages and lead to IL-1β release in response to *C. albicans* [8]. Apart from cell wall components, *C. albicans* can also release soluble factors that are able to trigger inflammasome activation. For example, the soluble factors, Sap2p, Sap6p and candidalysin, activate a cascade of events, such as potassium efflux, ROS production and lysosomal damage, which result in canonical NLRP3/caspase-1 pathway activation and, consequently, IL-1β/IL-18 production [8,30,39,85,86]. Of note, consistent NLRP3 activation, along with *CASP1* gene overexpression, as well as marked upregulation of the expression of some genes coding for important virulence factors in *C. albicans*, such as *SAP5*, *SAP6*, *HWP1* and *ECE1*, has been reported in women with VVC, but it has not been reported in asymptomatic carriers [40]. These findings provide evidence that NLRP3 inflammasome hyperactivation is a distinctive hallmark of VVC [40].

Inflammasome activation is typically considered to be beneficial to the host for mounting protective anti-*Candida* responses. In a murine model of VVC, Borghi et al. [87] demonstrated that during *Candida* infection, the NLRP3 inflammasome is negatively regulated by the interleukin (IL)-22/NLRC4/IL-1 receptor antagonist (IL-1Ra) axis. In their experimental model, IL-22, through the aryl hydrocarbon receptor (AhR), was found to play a key role in activating epithelial NLRC4, which in turn, restrained NLRP3 activity by inducing the production of IL-1Ra. Similarly, IL-22 has been shown to increase the expression of NLRC4, without affecting NLRP3 expression, in human vulvovaginal A431 cells in response to *C. albicans*. These data are supported by recent evidence for high levels of IL-1β and low levels of IL-1Ra and IL-22 in the vaginal fluids of patients with recurrent VVC; thus, dysregulation of the IL-22/NLRC4/IL-1Ra axis may contribute to VVC pathogenesis and lead to a chronic inflammatory state [8]. Consistent with this finding, infection in *NLRP3*-deficient mice or treatment of WT mice with the NLRP3 inhibitor, glyburide, reduced *C. albicans* vaginitis without affecting fungal colonization; this highlights the central role played by the NLRP3 inflammasome in *Candida* pathogenicity [88]. The NLRP3 inflammasome has also been implicated in the memory response termed “trained immunity”, which can occur both in mature myeloid cells, such as monocytes, DCs and natural killer cells, as well as in epithelial cells [89,90,91]. It is well established that this phenomenon is associated with epigenetic and metabolic reprogramming of innate immune cells [92,93].

Immune memory has been considered a unique feature of the adaptive immune response for a long time. In addition, there is evidence to indicate that the innate immune system may also result in enhanced responsiveness to subsequent stimuli. Although trained immunity is less specific and of a shorter duration than adaptive immune memory, its main function is to induce a quicker and stronger specific response against the same pathogen, as well as non-specific protection against subsequent challenges with the same or another pathogen. In past decades, trained immunity has been found to provide beneficial effects for the host by providing protection against pathogenic microorganisms. However, recent advances in research on this subject suggest that hyperactivation of the innate immune response, through trained immunity, could be associated with detrimental outcomes in patients affected by immune-mediated and chronic inflammatory diseases, such as VVC, and lead to pathological tissue damage [94,95]. In this context, dysregulation of the NLRP3 inflammasome pathway through massive recruitment of neutrophils, as an attempt to eliminate the fungal pathogen, further exacerbates vaginal inflammation. Of note, the inflammasome activation triggered by *C. albicans*, by leading to pyroptotic cell death and, consequently, macrophage lysis [96], may represent an immune evasion strategy adopted by the fungus to overcome the intracellular killing by phagocytic cells.

A gain-of-function mutation in the NLRP3 inflammasome has been found to be associated with inflammatory autoimmune diseases, and a growing body of evidence has revealed that polymorphisms and variable number tandem repeats in the *NLRP3* gene are associated with VVC [7,38]. Thus, considering the potential role of NLRP3 inflammasome hyperactivation in VVC pathogenesis, targeting the inflammasome pathways might provide novel and promising therapeutic approaches for downregulation of the inflammatory loop that promotes and exacerbates *Candida* pathogenicity [8]. In this respect, administration of anakinra, a recombinant interleukin-1 receptor antagonist (IL-1Ra), has been proven to reduce NLRP3-driven inflammation and protect against infection in a murine model of VVC [87]. Even though these findings highlight the potential benefits of inflammasome inhibitors in the treatment of VVC, future research in this direction is needed before such therapeutic approaches can be translated to clinical practice.

The role of neutrophils and macrophages in the immunophatogenesis of VVC is shown in Figure 2.

## 4. Relationship between Vaginal Microbial Communities and *C. albicans*

### 4.1. Vaginal Microbial Communities in Healthy Women

The human vagina is colonized by a variety of bacterial and fungal microbes that make up the normal microbiota and mycobiota, respectively. A growing body of evidence suggests that the microbiota composition and its functional properties play a fundamental role in the maintenance of vaginal health [97,98]. In healthy, reproductive-aged women, the vaginal microbiota is dominated by *Lactobacillus* spp., with *L. crispatus* being the major determinant of vaginal health [99]. Based on the abundance and composition of lactobacilli, vaginal bacteria species have been divided into diverse microbial community state types (CSTs) [100,101]. CST-I, CST-II, CST-III and CST-V are characterized by an abundance of *L. crispatus, L. gasseri*, *L. iners* and *L. jensenii*, respectively, while CST-IV includes low levels of lactobacilli and bacterial vaginosis-associated bacteria. Although CST-IV is frequently detected in healthy individuals, predominantly in black and Hispanic women (40%) [99], it is considered the most common dysbiosis state. It has been extensively demonstrated that lactobacilli, in particular *L. crispatus*, contribute to vaginal homeostasis by preventing *Candida* infection through different mechanisms (Figure 3).

On account of their ability to adhere to the vaginal mucosa, lactobacilli form a biological barrier and compete with *Candida* for adhesion sites on epithelial receptors. An in vitro study, investigating the ability of *L. crispatus* to adhere to VK2/E6E7 VECs and counteract *C. albicans* growth, showed that *L. crispatus* significantly inhibited the adhesion ability of *Candida* [102]. Moreover, lactobacilli produce a wide range of antimicrobial substances, such as lactic acid, bacteriocins, hydrogen peroxide (H_2_O_2_), and a variety of metabolites [103]. The production of a large amount of d-lactic acid by lactobacilli maintains the vaginal pH at <4.5 and inhibits the growth of potential pathogens. Furthermore, a low pH level impairs the dimorphic transition from the avirulent yeast to the virulent hyphal morphotype in *C. albicans* and, thus, contributes to maintaining the fungus in a state of immune tolerance [104]. Lactic acid, in both protonated l- and d-isomers, has also been shown to affect host immunity through diverse mechanisms, including its ability to dampen an exaggerated immune response by increasing the production of anti-inflammatory mediators, i.e., IL-1RA, from vaginal epithelial cells, as well as by inhibiting the production of pro-inflammatory cytokines, such as IL-6, IL-8, tumor necrosis factor alpha (TNF-α), RANTES (regulated on activation, normal T cell expressed and secreted) and macrophage inflammatory protein-3 alpha (MIP3α), by immune cells [105,106,107].

The production of H_2_O_2_ by *Lactobacillus* spp. may represent a nonspecific antimicrobial defense mechanism although, to date, its protective effect against *Candida* infection is debated. In vitro studies have shown that H_2_O_2_ exhibits a strong antifungal effect by inhibiting *Candida* overgrowth as well as its yeast-to-hyphal dimorphic transition [108,109,110,111]. In contrast, evidence makes its antimicrobial effect implausible in vivo [112]. Indeed, the human vaginal microenvironment is physiologically hypoxic, whereas lactobacillirequire oxygen to produce H_2_O_2_. Moreover, it has been demonstrated that in vivo levels of H_2_O_2_ are lower than those potentially microbicidal and that cervicovaginal fluid (CVF) and semen have significant H_2_O_2_-blocking activity [113,114].

Lactobacilli, especially *L. crispatus*, also produce biosurfactants that exert anti-*Candida* activity by reducing the adhesion of the fungus to HeLa cells and biofilm formation by different *Candida* species [99], and short-chain fatty acids (SCFAs), including acetic acid, propionic acid and butyric acid, which have been shown to have an anti-*Candida* effect due to their ability to inhibit not only *Candida* growth and germination but, also, the metabolic activity of *Candida* biofilm [14].

The exact mechanisms by which SCFAs may counteract *Candida* growth are not yet completely understood, even though several theories have implicated intracellular acidification, accumulation of anions, ATP depletion, and perturbation of the plasma membrane [115,116,117,118,119]. Further, functional perturbation of the plasma membrane may increase the uptake of antifungal drugs, such as azoles, and this may explain the synergistic effect of acetic/lactic acid and azoles on *Candida* growth [120]. In addition to SCFAs, lactobacilli also produce fatty acids with a chain length of 6–12 carbon atoms (MCFAs), i.e., caproic acid, heptanoic acid, caprylic acid, capric acid, undecanoic acid and lauric acid, which are known to have antifungal effects on *Candida* species [121,122,123,124,125,126]. Interestingly, *Saccharomyces boulardii,* a fungal yeast commonly used as a probiotic, produces caproic acid, caprylic acid and capric acid, among which capric acid has been identified as the most effective antifungal MCFA [122] capable of inhibiting the filamentation and adhesion of *C. albicans*. This explains the protective effect of *S. boulardii* against *C. albicans* infection. Finally, lactobacilli promote mucosal homeostasis by preserving the barrier function of the gut and vaginal epithelium through the production of certain metabolites. For instance, tryptophan catabolites, such as indole-3-aldehyde (3-IAld), an AhR ligand, act on regulatory T cells and lead to an increase in the local expression of IL-22, which in turn, plays an essential role in maintaining epithelial barrier integrity and function [127]. Indeed, while *L. reuteri* mainly contributes to preserving gut mucosal homeostasis via 3-IAld production [128], *L. acidophilus* has been found to exert a protective effect against VVC through the AhR/IL-22 axis [127]. Thus, the vaginal microbiota may mediate tolerance to *C. albicans* by providing colonization resistance to the fungus and mucosal protection from inflammation. However, *C. albicans*, by switching tryptophan degradation pathways from l-kynurenine to 5-hydroxyltryptophan pathways, can impair host defense mechanisms via suppressing the production of IL-17 [129]. These findings highlight the complex interaction between *Candida*, the host and the vaginal microbiota.

While we have extensive knowledge about the types of bacteria colonizing the vaginal microenvironment, to date, very little is known about their fungal counterparts. Research on this topic is still in its early stages, but recent studies have reported that in healthy women, *Ascomycota* is the dominant phylum, followed by *Basiodiomycota*. Within *Ascomycota*, the following orders have been identified: *Saccharomycetales*, *Capnodiales*, *Eurotiales*, *Pleosporales* and *Helotiales*. Without exception, the *Candida* genus is the predominant member of the fungal community, with *C. albicans* being the dominant species. Among non-*albicans* species, *C. glabrata* is the most common species. In addition, *C. krusei*, *C. tropicalis*, *C. parapsilosis*, *C. dubliniensis* and *C. guillermondii* have also been identified, with their prevalence varying according to the population studied and the methodology used. Other genera in the vaginal microenvironment include *Saccharomyces*, *Cladosporium*, *Aspergillus*, *Eurotium*, *Alternaria*, *Malassezia* and *Rhodotorula* [103,130,131,132]. The role of different fungal taxa in vaginal homeostasis has still not been explored. In the future, greater efforts are needed to analyse not only the composition of the vaginal mycobiota but also the intricate interplay of fungi with themselves, with the other microbes present in the vaginal milieu, and with their host. Understanding fungal taxa and their functionality is indispensable to elucidating the pathogenesis of VVC in order to design appropriate therapeutic interventions for the prevention and/or treatment of VVC.

### 4.2. Impact of Vaginal Dysbiosis on the Onset of VVC

In healthy women, vaginal homeostasis is maintained by complex and reciprocal interactions between the local microbial community and the host [133]. The vaginal microbiota undergoes dynamic changes during a woman’s life cycle. Various factors, i.e., hormonal changes, age, lifestyle, sexual practices, use of wide-spectrum antimicrobial drugs and ethnicity, may influence the composition of the vaginal microbiota [134].

Many studies have shown that vaginal dysbiosis is associated with various pregnancy complications, such as miscarriage, preterm birth and infertility. Vaginal microbiota, depleted of *Lactobacillus* spp. or other lactic acid bacteria (*Streptococcus, Enterococcus,* and *Atopobium*) and *Bifidobacterium*, has been associated with a poor reproductive outcome, including implantation failure and miscarriage [135,136]. Additionally, endometrial dysbiosis, with the dominant bacterial genus represented by *Burkholderia*, occurred in infertile women with repeated implantation failure [137].

Moreover, the general consensus is that fluctuations in the vaginal microbiota composition may favor the overgrowth of opportunistic pathogens and lead to the development of vaginal infections [138,139,140]. Until now, however, the role of vaginal bacterial communities in the development of VVC remains undefined, and the data reported in the literature are contradictory. Indeed, while it is known that vaginal dysbiosis increases the risk of sexually transmitted diseases [141,142], to date it does not seem to be associated with the risk of VVC/RVVC onset. In fact, vaginal microbiota in patients with VVC is mainly dominated by lactobacilli, as in healthy women [143,144].

However, even though the beneficial effects of lactobacilli on the vaginal ecosystem are well known, recent findings suggest that not all *Lactobacillus* spp. may have beneficial effects [98]. For example, while a *L. crispatus*-dominated vaginal microbiota (CST I) is always associated with a healthy vagina, the predominance of *L. iners* in the vaginal milieu (CST III) is more likely to be associated with vaginal dysbiosis and to predispose individuals to both bacterial vaginosis (BV) and *Candida* infections [145].

An increasing number of studies have shown that *L. iners,* an unusual *Lactobacillus* species, offers lesser protection against vaginal colonization by potential harmful pathogens than other known beneficial lactobacilli, because it is unable to produce d-lactic acid (and able to produce only l-lactic acid). The d-lactic acid isomer has been proven to have greater inhibitory activity against exogenous bacteria than l-lactic acid [146]. This might explain why *L. iners* is less effective in counteracting invading pathogens. Furthermore, as it is almost completely unable to produce d-lactic acid, it is associated with a high l/d-lactic acid ratio. This may induce an increase in the level of extracellular matrix metalloproteinase inducer and lead to the activation of matrix metalloproteinase-8, which can cause the breakdown of the extracellular matrix, thus favoring pathogen penetration and, subsequently, infections of the upper genital tract [147]. In contrast, *L. iners* is known to produce inerolysin, a cholesterol-dependent cytolytic toxin, which has the potential to damage VECs. The *L. iners*-associated cytolysin is thought to be similar to vaginolysin and is expressed by several BV-associated bacteria [147,148]. Moreover, the expression of cytolysin secreted by *L. iners* is highly upregulated in a state of vaginal dysbiosis compared to balanced microbial environments [149]. On the other hand, an interesting study conducted by Zhou et al. demonstrated that metronidazole treatment of women affected by BV was associated with a *L. iners*-dominated vaginal microbiome, and in vitro experiments also validated the inhibitory effect of *L. iners* on anaerobic bacteria, such as *Gardnerella vaginalis* and *Fannyhessae vaginae* [150]. The findings of this study suggest that *L. iners* is associated with positive outcomes after BV treatment; thus, its antimicrobial effect could be beneficial to patients with BV [150]. In addition to the suspected role of *L. iners* in the onset of BV, mounting evidence also implicates its role in the pathogenesis of VVC. In this regard, Tortelli et al. [151] reported that *L. iners*-dominant communities were more likely to harbour *Candida* than non-*L. iners*-dominant communities. Moreover, Ceccarani et al. showed that VVC-positive women exhibited an increase in the relative abundance of *L. iners*, along with a decrease in the abundance of *L. crispatus*, and Novak et al. demonstrated that *L. iners*-dominated CST III vaginal microbiota did not provide optimal protection against *Candida* spp. colonization [152,153].

It is well known that the ability of *C. albicans* to produce biofilms is one of the most important determinants of vaginal candidiasis, since it acts as a physical barrier that confers resistance to conventional antimycotic drugs, i.e., fluconazole and amphotericin B [154,155,156,157]. Various *Lactobacillus* species, such as *L. rhamnosus*, have been shown to inhibit *C. albicans* biofilm production and biofilm-related gene expression when co-cultured [158]. In addition, biosurfactants produced by *L. crispatus* were found to reduce the adhesion of *C. albicans* to HeLa cells and biofilm formation, and biosurfactants produced by *L. gasseri* and *L*. *jensenii* were able to inhibit biofilm production by several species of *Candida*, including *C. albicans, C. tropicalis* and *C. krusei* [159]. Conversely, several studies have demonstrated that *L. iners* enhances the ability of *Candida* to produce biofilms. For example, McKloud et al. demonstrated that co-culture of *C. albicans* with *L. iners* resulted in upregulation of the expression of biofilm-related genes, particularly *ALS3* and *ECE1* [158]. Our recent findings have further strengthened the evidence in favor of the role of *L. iners* in the pathogenesis of VVC. The findings indicated that a *L. iners*-dominated vaginal microbiome may not only contribute to the onset of VVC, but also increase the probability of RVVC development. Notably, we demonstrated that *L. iners* cell-free supernatants were able to potentiate the virulence of 77.7% of the *C. albicans* clinical isolates tested by inducing a significant increase in both biofilm biomass and metabolic activity and transforming *Candida* vaginal isolates from moderate or weak biofilm-producing strains to strong biofilm producers. This effect was associated with enhancement of hyphal/pseudohyphal growth and high-level expression of *HWP1* and *ECE1*, which are typical hypha-associated genes [160]. It is well established that the vaginal microbiota balance affects local immune homeostasis. In this regard, it has been reported that, unlike *L. crispatus*-dominated microbiota, vaginal microbiota dominated by *L. iners* are associated with high levels of pro-inflammatory factors, including IL-18, IL-1α and macrophage migration inhibitory factor, which are known to activate T cell proliferation, as well as high levels of IL-12p70 (which is responsible for the increased production of TNF-α). Thus, a vaginal mucosal surface, persistently colonized by *L. iners*-dominated microbiota, could favor, in the long term, the establishment of a comfortable environment for the development of infections characterized by clinically overt inflammation, as observed in the case of vaginal candidiasis (Figure 4).

Based on the research on *L. iners* reported so far, more studies on this *Lactobacillus* species are warranted to clarify the precise role of this intriguing species in the onset and recurrence of VVC. Overall, these findings suggest that, contrary to other lactobacilli, *L. iners* has a complex and ambiguous role in the vagina. Therefore, considering that the presence of *L. iners* could be indicative of a state of vaginal dysbiosis, *L. iners*-based probiotics should be avoided as a therapeutic option for the treatment of VVC.

## 5. *C. albicans*-Related Key Virulence Factors in the Pathogenesis of VVC/RVVC

### 5.1. Morphological Transition of Candida

Even though the immunopathogenic host response characterized by NLP3 inflammasome hyper-activation, both in epithelial and myeloid cells, is considered to be the key player in the onset of VVC, the virulence factors that govern *C. albicans* pathogenesis are equally important in influencing the disease process. Apart from adhesins, which mediate the first step of *Candida* infection, that is, the attachment of the fungus to VECs, and are also involved in biofilm formation, other virulence factors are involved in the spread of the infection. For instance, the ability of *C. albicans* to undergo morphological transition from yeast, into true hyphae or pseudo-hyphae, is considered to be one of its major virulence traits, as this facilitates tissue invasion and also provides resistance against phagocytosis [161]. The dimorphic transition contributes to *Candida* virulence by playing a role in overcoming the epithelial tolerance threshold [10]. Moreover, the yeast-to-hyphal transition allows for the rapid adaptation of *Candida* to various stressful environments and the remodulation of the cell surface antigenic composition, as a result of which, it cannot be promptly recognized by previously sensitized immune cells. Thus, specific episodes of RVVC may occur in vaginal microenvironments colonized by *Candida* strains that intrinsically change or modulate their antigens, and this may result in more virulent strains, and, therefore, influence the transition from commensalism to pathogenicity.

### 5.2. Secretory Factors

*C. albicans* produces several virulence factors, such as candidalysin and hydrolytic enzymes, which facilitate tissue invasion, immune stimulation and evasion, organ damage and nutrient acquisition.

Candidalysin is a cytolysin belonging to candidalysins, a new family of fungal peptide toxins [37]. It was the first cytolytic peptide toxin to be identified in human fungal pathogens [35]. This toxin is produced by targeted proteolytic processing of the hypha-associated protein, Ece1p and, following secretion, damages epithelial cells and induces immune signaling in them, primarily by EGFR activation [162]. Its crucial role in VVC immunopathology was clearly demonstrated in mice intravaginally challenged with *C. albicans* strains deficient in candidalysin that did not differ in the ability to undergo yeast-hypha phenotypic switching but induced a significant decrease in neutrophil recruitment and tissue damage compared to isogenic controls [36].

As demonstrated by recent studies, the toxin ability to form pores in the membrane of epithelial cells is crucial not only for its cytolytic effect, but also for immunostimulatory activity [162,163,164]. Indeed, by forming pores, candidalysin activates the release of pro-inflammatory cytokines and the recruitment of immune cells through complex intracellular pathways involving both EGFR-dependent and EGFR-independent signaling [165].

As previously mentioned, the activation of NLRP3 inflammasome is another important immunostimulatory property of candidalysin. Several works [166,167,168] showed, both in murine and human mononuclear phagocytes, and also in the human macrophage-like THP1 cell line, that candidalysin is necessary and sufficient for activating NLRP3, leading to IL-1β production. Of note, Roselletti et al. [30,40] have recently demonstrated that women with VVC exhibited a consistent overexpression of the NLRP3 inflammasome and CASP1 genes in vaginal samples, with respect to colonized individuals. In addition, the inflammasome expression was coupled with neutrophil recruitment and IL-1β and IL-8 production.

Intriguingly, recently [169], different candidalysin isoforms (SC5314- or 529L) have been reported and, specifically, the 529L isoform was associated with a reduced vaginal pathogenicity in a mouse model of VVC, due to reduced capacity to be processed and secreted by *C. albicans*. In a more recent study, the candidalysin alleles, SC5314 or 529L-like, have been found in vaginal clinical isolates, but no association was observed between the two candidalysin isoforms and symptomatic VVC [44]. In light of these data, a deeper polymorphism analysis of virulence-related factors in *C. albicans* is needed for a better understanding of the differences in *Candida* pathogenicity.

Interestingly, candidalysin has also been recently identified as a hemolytic factor of *C. albicans* [170]. Accordingly, candidalysin may be responsible for producing hemocidins in the vaginal environment, through the lysis of RBCs present in menstrual discharge that are generated proteolytically from hemoglobin. Hemocidins are, in fact, a group of microbicidal peptides arising from heme-binding proteins such as hemoglobin [171]. Given that such peptides can be found in menstrual discharge [172], they could strongly influence the composition of the vaginal microbiota and, thus, contribute to excessive growth of *C. albicans* and favor vaginal candidiasis. It is known that hemolysins, such as the α-hemolysin of *Escherichia coli* and the α-toxin of *Staphylococcus aureus*, induce erythrocyte lysis via activation of purinergic P2 receptors present on the cell surface. Therefore, the purinergic receptor P2X antagonist pyridoxal-phosphate-6-azophenyl-2′,4′-disulfonic acid or anti-candidalysin nanobodies could be promising therapeutic agents for combating *Candida* infections [170].

Various extracellular hydrolases, including aspartyl proteases, lipases and phospholipases, secreted by *Candida*, play an important role in its pathogenicity. In fact, these enzymes are associated with adhesion, invasion and tissue damage that result from derangement of the constituents of the host cell membrane [173,174]. Of these, aspartyl proteases, SAP2 in particular, are involved in the degradation of human proteins, such as mucin and mucosal secretory immunoglobulin A; these effects favor *Candida* adhesion and tissue penetration by eliminating the protective host barrier at the mucosal level [175]. SAP1-3 promotes immune evasion by degradation of the complement components, C3b, C4b and C5, and, therefore, reduces the inhibitory activity of the complement system [176]. In addition, it has been demonstrated that SAP2 and SAP6 are able to recruit neutrophils, both directly and indirectly, through activation of the NLRP3 inflammasome and induction of the pro-inflammatory cytokines IL-1β and IL-18, and chemokines, such as MIP-2, in the early stage of vaginal inflammation [39,41]. On the other hand, Bruno et al. implicated SAP5, a typically hypha-associated enzyme, in the late immune-pathogenic activity of *C. albicans* [38]. The mechanisms by which neutrophils sense SAPs remain unclear. Intriguingly, it has been demonstrated that the proteinase content in vaginal strains isolated from women with acute vaginal candidiasis was significantly higher than that of strains isolated from asymptomatic vaginal carriers [177]. Given the central role played by SAPs in the pathogenesis of VVC, they could be important for the development of targeted vaccines and therapies. In fact, a SAP2-based virosomal vaccine (PEV7) has been developed and found to provide efficient protection against vaginal candidiasis in animal models [178].

Recently, it was confirmed that *Candida* phospholipase contributes to the pathogenic process of VVC by affecting the integrity of the cell membrane, through the decomposition of host cell membrane phospholipids, and thereby, promoting tissue invasion [27]. Thus, it is conceivable that *Candida* strains able to produce large amounts of proteinases and phospholipases may strongly influence host susceptibility to VVC.

### 5.3. Biofilm Formation

Apart from the wide array of virulence factors produced by *C. albicans*, biofilm production is another strategy by which this pathogen initiates and perpetuates infection. Biofilms are microbial communities attached to abiotic or biotic surfaces, such as medical devices and host tissues during infection; the microbial communities are encased in a self-produced extracellular polymeric matrix composed of polysaccharides, proteins, DNA and lipids. Unlike other *Candida* spp., *C. albicans* biofilms display a more complex and heterogeneous organization: it has a multi-layer structure composed of yeast cells, pseudo-hyphae and true hyphae, surrounded by an extracellular matrix [179]. In *C. albicans*, biofilm formation is strictly dependent on the ability of the fungus to switch from the yeast form to the mycelial form. *C. albicans* mutants, which exhibit defects in hyphal formation under in vitro conditions, were found to have defects in biofilm production as well [180,181].

Among *Candida* surface proteins, Hwp1 is a well-characterized *C. albicans* cell surface adhesin that is expressed only on the hyphal morphotypes, and is involved in biofilm formation at the mucosal level, by mediating the binding of the fungus to epithelial cells [180]. Epithelium-associated biotic biofilms may represent a persistent source of infection for patients, and may play an important role in the pathogenesis of RVVC. Indeed, mature biofilms can continuously release *Candida* cells in the vaginal microenvironment and, thus, lead to colonization of new sites and even chronic infection in some cases. Furthermore, fungal cells encased in biofilms exhibit different phenotypes compared to their planktonic counterparts, i.e., lower growth, especially those within deeper layers of the mature biofilm, and high resistance to host immune defense mechanisms as well as conventional antifungal agents, and this often leads to the failure of therapeutic treatments.

Preclinical studies in a mouse model of vaginal candidiasis highlighted the ability of *C. albicans* (reference strain SC5314 and clinical isolates) to successfully establish biofilms on the vaginal epithelium and proposed that the formation of epithelium-associated biofilms in the vagina may be the initial event in the establishment of vaginal infection by *C. albicans* [154,182]. Therefore, it is quite likely that biofilm formation, both on the vaginal mucosa and on abiotic surfaces, such as intrauterine devices, may represent a potential risk factor for RVVC [183].

Recently, several works [158,160,184] reported that *C. albicans* clinical isolates, including the vaginal ones, display a high degree of heterogeneity, with respect to their capacity to form biofilms, which correlates with their pathogenicity and altered antifungal drug sensitivity [184,185]. These results, therefore, suggest that biofilm heterogeneity could affect the management of *C. albicans* vaginal infections, and highlight the need to further characterize the phenotype and the antifungal sensitivity profile of biofilms.

The management of biofilm-associated *Candida* infections is even more difficult due to the ability of *C. albicans* to form polymicrobial biofilms. Indeed, it is the most common fungal pathogen frequently co-isolated from polymicrobial biofilms in different human body sites, including the vaginal environment. This is due to the fact that *Candida* hyphae can act as a potential bio-scaffold for the colonization of its neighboring bacteria [186,187,188]. It has been reported that approximately 20–34% of RVVC samples contain vaginal bacterial pathogens such as *Streptococcus agalactiae* (group B *Streptococcus*, GBS) and *Gardnerella vaginalis* [189]. A survey on biofilms formed on intrauterine devices confirmed the presence of *C. albicans* together with numerous aerobic and anaerobic bacterial pathogens [190]. As demonstrated in several studies, the mixed microbial community are more resistant to antimicrobial agents, as compared to mono-species biofilms [191]. For instance, the antifungal effect of miconazole is reduced in a dual-species *C. albicans-Staphylococcus aureus* biofilm [186].

Overall, deciphering the molecular mechanisms of the interactions between *C. albicans* and the resident microbiota may be of great importance in designing therapeutic strategies aimed to combat polymicrobial biofilm infections.

### 5.4. Potential Role of β-Glucan Unmasking in the Pathogenesis of VVC

Shielding the immunogenic cell wall epitope β-(1,3)-glucan, under an outer layer of mannosylated glycoproteins, is another key virulence factor, adopted by *C. albicans* during systemic infections, to reduce visibility to the immune system. As aforementioned, the antigenic cell wall polysaccharide, β-(1,3)-glucan, is detected by the innate immunity cells through the Dectin 1 receptor. This recognition can trigger phagocytosis of fungal pathogens and protective antifungal immune responses in innate immune cells like macrophages, dendritic cells and neutrophils. However, under normal conditions, *C. albicans* masks β-(1,3)-glucan from immune detection via the outer layer of mannosylated proteins. Unmasking of β-(1,3)-glucan can be induced by treatments with drug or genetic mutations that disrupt the integrity of the cell wall [192,193].

Interestingly, studies, performed in murine models of both oropharyngeal (OPC) and vaginal candidiasis, found that mutants with increased β-(1,3)-glucan exposure (unmasking) displayed increased immunostimulatory capabilities in vitro, and attenuated virulence in the OPC model [187]. In contrast, the unmasking mutants failed to decrease the fungal burden during VVC, although they were able to induce high levels of polymorphonuclear cells and interleukin-1β (IL-1β) within the vaginal lumen [194].This suggests that the unmasked cells do not provide a beneficial outcome due to their ability to stimulate an immunopathogenic microenvironment during vaginal candidiasis. This is in agreement with a previous study, as discussed above, showing that heparan sulfate within the vaginal lumen competes with the neutrophil receptor, Mac-1, to impair the recognition of fungal cells by neutrophils [69].

Diversity in cell wall composition, specifically in chitin and β-glucan exposure, have been observed in clinical vaginal isolates. Particularly, Pericolini et al., demonstrated that the β-glucan was largely masked from the immune system, especially on yeast, and only exposed on a small percentage of hyphae, and that enhanced β-glucan visibility was found in symptomatic women with strong neutrophil infiltration [195].

In another study, Gerwien et al. [196] demonstrated that vaginal clinical isolates from asymptomatic or symptomatic women with VVC, when cultured in a laboratory-rich medium (YPD), exhibited significantly less β-glucan but higher mannan and chitin content than the laboratory strain SC5314. On the contrary, in a vaginal-simulating medium, the isolates from asymptomatic women displayed less chitin, while those from women with VVC displayed more β-glucan, compared to the SC5314 strain [196]. These findings highlight that diversity in cell wall architecture in *C. albicans* may drive an altered initiation of an immune response in the corresponding host microenvironment, explaining the differences in VVC pathogenicity.

## 6. Potential Therapeutic Strategies for the Prevention and/or Treatment of VVC

Various therapeutic strategies have been employed in the clinical setting for the treatment of VVC. Fluconazole, amphotericin B, nystatin and flucytosine are the most common antifungal drugs currently in use to treat *Candida* vaginitis. To date, fluconazole, in the form of either topical formulations or a single oral dose, remains the first-line drug for the management of uncomplicated VVC. However, it is noteworthy that 40% to 50% of women treated for RVVC will experience recolonization with *Candida* within 30 days after therapy cessation [197], and that prolonged treatment with antifungal agents may induce drug resistance in *Candida* [198]. Thus, there is a substantial need for the development of novel therapeutic strategies for the prevention and treatment of VVC/RVVC.

As vaginal candidiasis is now considered an immunopathological condition, therapeutic approaches focus on preventing and/or reducing the host hyper-inflammatory response to *C. albicans*. Thus, adjuvant immunotherapies, by targeting hyper-inflammation, might represent a therapeutic option to ameliorate the inflammatory immune response and alleviate symptoms in patients affected by acute VVC. To this end, given the key role that leukotrienes—a group of inflammatory vasoactive mediators—play in promoting inflammation, leukotrien receptor antagonists (LTRAs) have attracted great interest. However, conflicting data have been reported in the literature, regarding their efficacy in the treatment of VVC. For example, a study showed that the LTRA, zafirlukast, provided symptom relief in women with RVVC [199], but in a murine model of VVC, the LTRAs, montelukast and zafirlukasts, did not affect the production of inflammatory mediators and vaginitis-associated tissue damage in the vagina [200]. Moreover, as discussed earlier, inhibitors of the NLRP3 inflammasome or IL-1Ra inhibitors—such as anakinra—as well as AhR agonists, have been proposed as potential therapeutic agents, but further studies are necessary to demonstrate their efficacy [87,201,202].

More recently, it has been discovered that retinoids—in particular the all-trans retinoic acid (ATRA), which is an active metabolite of vitamin A—have potential beneficial effects in the treatment of fungal infections [203,204] by exerting a dose-dependent inhibitory effect against *C. albicans* growth, hyphal development and biofilm formation under in vitro conditions [205]. Therefore, on account of its antifungal properties, ATRA could be a potential candidate in the treatment of VVC.

Recently, numerous studies have indicated that lactobacilli-based approaches may be a promising alternative to classical antifungal therapies for the prevention and/or treatment of RVVC. Targeting vaginal dysbiosis, by replacing beneficial bacteria, has been shown to improve the clinical outcomes of women affected by VVC. For example, oral administration of the specific probiotic strain, *Lactobacillus plantarum* P17630, was associated with lactobacilli colonization on VECs and improvement of clinical signs, such as redness, swelling and discharge, in women with a history of RVVC [206]. Further, long-term administration of lactobacilli-based probiotics, in both experimental animal models and humans, has proven to be effective for the prophylactic treatment of RVVC. Specifically, *L. crispatus* and *L. delbrueckii* were found to inhibit 60–70% of *C. albicans* in a Sprague Dawley rat model of VVC [207]. In addition, the effectiveness of oral or intravaginal administration of *L. acidophilus*, *L. rhamnosus* GR-1 and *L. fermentum* RC-14, in colonizing the vagina and/or preventing vaginal colonization and infection by *C. albicans,* has been demonstrated in clinical trials [208]. Furthermore, lactobacilli have exhibited beneficial effects in the prevention and treatment of RVVC when administered along with standard antifungal drugs [209].

In recent years, yeast-based probiotics have been proposed as a promising tool for the prophylaxis and treatment of vaginal infections. Preclinical in vitro and in vivo studies have identified various yeast-based probiotics that are able to counteract the virulence of *Candida* spp. However, to date, *S. cerevisiae* var. *boulardii* is the only probiotic yeast commercially available for human use. In a mouse model of VVC, intravaginal administration of live or inactivated *S. cerevisiae* (CNCM I-3856 strain) was found to promote *C. albicans* clearance at levels similar to those achieved with fluconazole; this effect was the result of the ability of *S. cerevisiae* to co-aggregate with *Candida* cells, thus preventing the adhesion of the fungus to the vaginal epithelium [210,211]. Moreover, this strain was able to negatively affect *C. albicans* virulence factors, such as dimorphic transition from the yeast to hyphal form, by inhibiting the expression of the hyphal growth-associated genes, i.e., *HWP1* and *ECE1*, as well as those encoding for secretory aspartyl proteinases [210,212]. The therapeutic activity of live *S. cerevisiae* CNCM I-3856 has also been demonstrated in clinical studies, and its recovery in vaginal samples after oral administration suggests that this strain might have great potential for controlling *C. albicans* load in the vagina, and also for preventing VVC recurrence in women receiving conventional antifungal drugs for the treatment of VVC [213].

Based on the success of fecal microbiota transplantation, as a biotherapeutic treatment for gut diseases such as *Clostridium difficile* infections, VMT from healthy women has been proposed as a therapeutic option for the treatment of vaginal dysbiosis. VMT has demonstrated therapeutic efficacy in rat models of vaginal dysbiosis by reducing inflammation, restoring vaginal microbiota balance, increasing the number of lactobacilli and reducing endometritis-like symptoms [214,215]. The therapeutic benefits of VMT have also been proven in patients with symptomatic, intractable and recurrent bacterial vaginosis, and this has opened up a new avenue for such future studies [216]. However, the potential use of VMT, in a clinical setting for the treatment of gynecological infectious diseases, is associated with many issues, such as insufficient VMT clinical trials (only one trial with five subjects) [216], a lack of standard guidelines, potential transmission of drug-resistant microbes, immune rejection and lack of information about long-term effects. Therefore, a multidisciplinary approach is required to develop a new safe and effective VMT-based protocol for the treatment of BV and other gynecological diseases such as VVC. The aforementioned potential therapeutic strategies are summarized in Table 1.

## 7. Conclusions and Perspectives

In conclusion, it is clear that vaginal health is dependent on a complex and functional equilibrium between the host immune response, vaginal microbiota and *C. albicans*. Numerous factors that contribute to the variability of individuals and their microbiota, other than genetic and phenotypic variation among clinical isolates of *C. albicans*, further increase this immense complexity. Specifically, host-related factors, such as a dysregulated immune response triggered by *C. albicans* or an altered vaginal microbiota composition, may substantially contribute to individual variability, predisposing women to VVC.

There is now a general consensus that VVC is considered an immunopathology, in which both fungal virulence factors, such as the hyphal form and candidalysin, as well as host immune response, like genetic polymorphisms of the NLRP3 inflammasome, are all important actors in driving hyper-inflammation. In this context, a deeper understanding of the molecular mechanisms underlying the immune hyper-inflammatory response towards *C. albicans* could help to identify specific targets for novel therapeutic approaches in the treatment of vaginal candidiasis.

Disruption of the vaginal ecosystem may also contribute to the overgrowth of pathogens that are responsible for complicated vaginal infections, including VVC. In particular, a reduction in specific health-associated *Lactobacillus* spp., such as *L. crispatus*, coupled with an increase in *L. iners*, can predispose women to *C. albicans* infection. The mechanisms by which *L. iners* offers less protection against the fungus, leading to the onset of VVC, are not yet completely understood. To this end, further studies are required to better characterize the exact role of this intriguing species in vaginal health and diseases, such as vaginal candidiasis. Understanding of the complexity of the interaction between *C. albicans* and vaginal lactobacilli could offer potential avenues for improving women’s health through probiotic-based therapeutic approaches. However, it should be noted that although many in vivo and in vitro studies have demonstrated that *Lactobacillus* spp. have a beneficial effect on the prevention and/or treatment of VVC, clinical data are still scarce and need to be further implemented.

Last but not least, phenotypic and genotypic variations within *C. albicans* may lead to the development of more virulent strains that might influence vaginal candidiasis by altering the balance between *C. albicans* commensalism and pathogenicity. In this respect, it must be pointed out that most studies on VVC have been conducted in experimental models by using laboratory reference strains of *C. albicans*, which could not mirror the real virulence and pathogenicity of the clinical isolates. Given these limitations, future research should focus on the clinical isolates in order to simulate the real disease. Furthermore, deciphering the molecular mechanisms regulating strain virulence may be relevant to identify potential therapeutic fungal targets for the treatment of this infection as well.

It is also noteworthy that mouse models have been used extensively to study *C. albicans* pathogenicity during VVC. However, although rodent models show some similarities with the immunological and physiological properties of human VVC, they might not reflect human infection properly. The main differences consist in a vaginal pH that is neutral in mice, and the fact that *C. albicans* is not a normal commensal of the vaginal microbiota of rodents. Therefore, selecting appropriate models may be helpful to better study the pathophysiology of VVC. As future perspectives, the use of human vaginal organoids, which can recreate the architecture and physiology of the vagina, might provide an appropriate and valid system for a deeper comprehension of the complex interplay between host response and *C. albicans*.

In summary, despite significant progress in our understanding of *C. albicans* as a commensal, and the factors that prompt the fungal transition from commensal to pathogen, many aspects of vaginal candidiasis are still not fully understood.

## Figures and Tables

**Figure 1 microorganisms-11-01211-f001:**
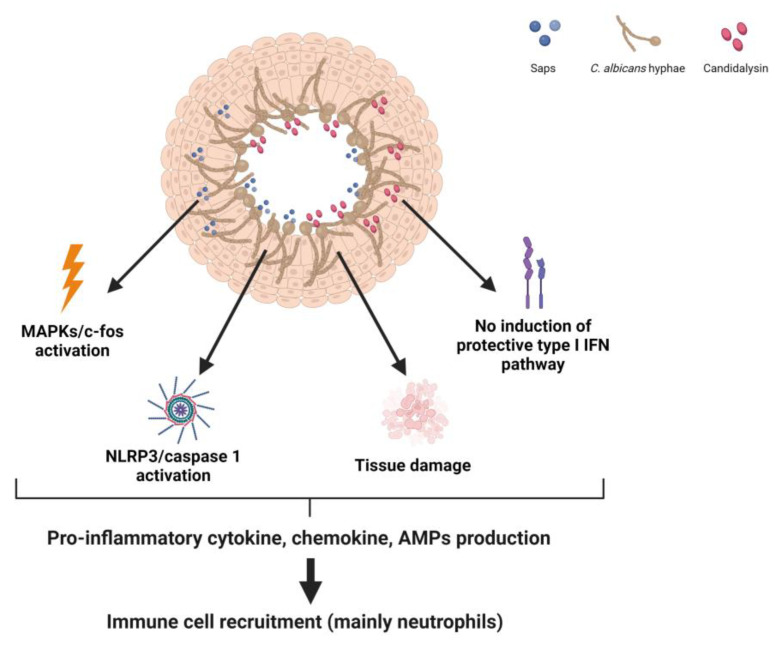
Schematic representation of the principal effects resulting from the *C. albicans* hyphae-VECs interaction in VVC. The hyphal form of *C. albicans* and hyphae-associated virulence factors, such as candidalysin and SAPs, induce tissue damage and trigger a strong and sustained activation of VECs, leading to the release of inflammatory mediators such as cytokines and chemokines, (i.e., IL-1α, IL-1β, G-CSF, IL-6, GM-CSF, IL-36γ and CCL20) and antimicrobial peptides (AMPs; alarmins, beta-defensins) via MAPKs/c-fos signaling. Candidalysin and SAPs also trigger NLRP3 inflammasome, which in turn, activates the production of IL-1β. All these inflammatory mediators drive the vaginal recruitment of a large number of immune cells, primarily neutrophils, which are unable to internalize and kill the fungus, perpetuating the inflammatory loop. In addition, hyphal cells fail to elicit a type I IFN response that protects VECs against *Candida*-induced damage. Created with BioRender.com.

**Figure 2 microorganisms-11-01211-f002:**
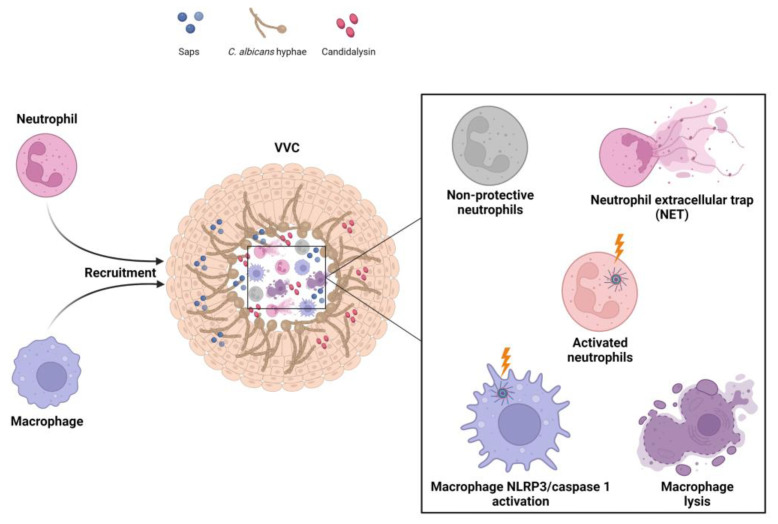
Role of neutrophils and macrophages in the immunophatogenesis of VVC. Neutrophils and macrophages, recruited into the vaginal epithelium, contribute substantially to hyper-inflammation through different mechanisms. Indeed, despite a massive infiltration, neutrophils are unable to efficiently kill *Candida* due to the presence of high concentrations of inhibitors (i.e., heparan sulfate and pANCA) in the vaginal environment. In addition, NET formation leads to neutrophil death and, consequently, to the release of their intracellular contents, thus enhancing the tissue damage. The NLRP3/caspase 1 pathway activation in neutrophils and macrophages results in inflammatory cytokine production (IL-1β and IL-18) that contributes to the amplification inflammation. Furthermore, candidalysin promotes *C. albicans’* escape from macrophage killing by permeabilizing macrophage cell membrane and engaging two host cell death pathways, namely, Gasdermin D-mediated pyroptosis and Etosis. The release of inflammatory mediators from dying macrophages further exacerbates the inflammatory status and fungal-mediated tissue damage. Overall, neutrophils and macrophages are not only unable to clear the fungal cells, but also contribute to the progression of inflammation and tissue damage triggered by *C. albicans.* Created with BioRender.com.

**Figure 3 microorganisms-11-01211-f003:**
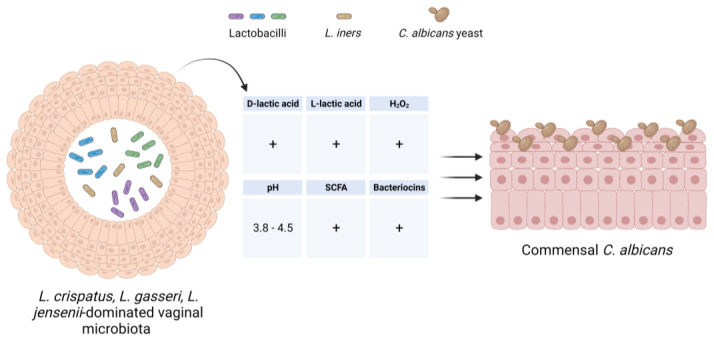
*Lactobacillus* spp. in a healthy vaginal environment. The vaginal microbiota of healthy, reproductive-age women is mainly dominated by lactobacilli belonging to CST-I, CST-II and CST-V, with *L. crispatus* being the major determinant of vaginal health. Lactobacilli contribute to maintaining the vaginal homeostasis by different mechanisms. They compete with *C. albicans* for adhesion to the surface of vaginal epithelium. Moreover, lactobacilli counteract the overgrowth and yeast-to-hyphal dimorphic transition of *C. albicans* by the release of a broad spectrum of antifungal compounds, including lactic acid, H_2_O_2_, SCFAs and bacteriocins. SCFA: short chain fatty acids. Created with BioRender.com.

**Figure 4 microorganisms-11-01211-f004:**
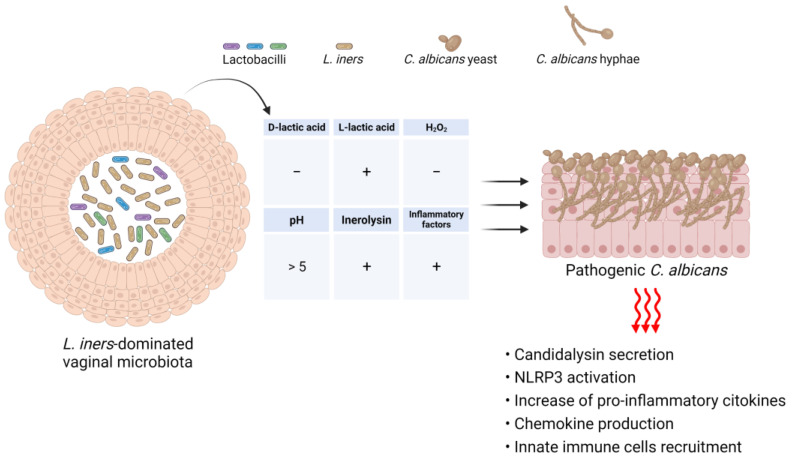
Potential role of *Lactobacillus iners* in the pathogenesis of VVC. Vaginal microbiota dominated by *Lactobacillus iners* induces a pro-inflammatory environment and the commensal-pathogen transition of *C. albicans* by the enhancement of filamentation and biofilm production. VVC onset is triggered by the secretion of *C. albicans* cytotoxic proteins and the activation of the NLRP3 inflammasome pathway, which leads to cytokine and chemokine production by epithelial cells. The massive recruitment of innate immune cells in the vaginal canal is the leading cause of the hyper-inflammatory state during *C. albicans* infection. Created with BioRender.com.

**Table 1 microorganisms-11-01211-t001:** Potential therapeutic strategies for the prevention and/or treatment of VVC/RVVC.

Potential Therapeutic Strategies	Effects	References
**Immunomodulators**		
Zafirlukast (LTRA)	Symptom relief in women with RVVC	[199]
Anakinra (IL-1Ra), Indole-3-Aldehyde (AhR agonist)	Reduction of inflammation in murine models of VVC	[87,202]
ATRA	Fungistatic effect against *C. albicans* growth and biofilm formation in in vitro model	[205]
**Lactobacilli-based probiotics**		
*L. plantarum* P17630	Vaginal colonization by lactobacilli and improvement of clinical signs in women with RVVC	[206]
*L. crispatus* *L. delbrueckii*	Antifungal efficacy in a rat model of VVC	[207]
*L. acidophilus * *L. rhamnosus* GR-1 *L. fermentum* RC-14	Effectiveness in preventing *C. albicans* colonization and infection in women with RVVC	[208,209]
**Yeast-based probiotics**		
*S. cerevisiae* (CNCM I-3856 strain)	Reduction of *C. albicans* load and vaginal inflammation in a murine model of VVC;	[210,211,212]
	Colonization of human vagina after oral administration	[213]
**VMT**	Therapeutic efficacy in rat models of vaginal dysbiosis;	[214]
	Reduction of endometriosis disease progression in female mice;	[215]
	Improvement of symptoms in patients with RBV	[216]

IL-1Ra: Recombinant interleukin-1 receptor antagonist; LTRA: Leukotriene receptor antagonist; AhR: Aryl hydrocarbon Receptor; ATRA: All trans retinoic acid; VMT: Vaginal microbiota transplantation; RBV: Recurrent bacterial vaginosis.

## Data Availability

No new data were created in this study. All the data reported in this review were found in original articles cited in the text. Literature used to inform the text of this article was selected from PubMed.gov from the National Library of Medicine.

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
