# Peer review of "The Interplay between Candida albicans, Vaginal Mucosa, Host Immunity and Resident Microbiota in Health and Disease: An Overview and Future Perspectives"

_microorganisms, 2023, doi:10.3390/microorganisms11051211_

Round 1

Reviewer 1 Report

microorganisms-2304987

Review Comment

This manuscript is well- designed-and-written.

I have only one suggestion.

Vaginal dysbiosis is thought to be associated female infertility.

I recommend additional discussion on this matter and quotation of the following articles investigating vaginal microbiota in infertile women:

Kitaya, K.; Nagai, Y.; Arai, W.; Sakuraba, Y.; Ishikawa, T. Characterization of microbiota in endometrial fluid and vaginal secretions in infertile women with repeated implantation failure. Mediat. Inflamm. 2019, 2019, 4893437.

Tanaka, S.E.; Sakuraba, Y.; Kitaya, K.; Ishikawa, T. Differential Vaginal Microbiota Profiling in Lactic-Acid-Producing Bacteria between Infertile Women with and without Chronic Endometritis. Diagnostics 2022, 12, 878.

Author Response

Dear reviewer,

thank you very much for your review and valuable suggestion.  The changes within the text are highlighted in green.

R.: Vaginal dysbiosis is thought to be associated with female infertility.

I recommend additional discussion on this matter and quotation of the following articles investigating vaginal microbiota in infertile women:

Kitaya, K.; Nagai, Y.; Arai, W.; Sakuraba, Y.; Ishikawa, T. Characterization of microbiota in endometrial fluid and vaginal secretions in infertile women with repeated implantation failure. Mediat. Inflamm. 2019, 2019, 4893437.

Tanaka, S.E.; Sakuraba, Y.; Kitaya, K.; Ishikawa, T. Differential Vaginal Microbiota Profiling in Lactic-Acid-Producing Bacteria between Infertile Women with and without Chronic Endometritis. Diagnostics 2022, 12, 878.

A.: As suggested, we have discussed the impact of vaginal dysbiosis on female fertility by referring to the articles by Kitaya et al. and Tanaka et al.”. Please see paragraph 4.2 lines 432-438.

Reviewer 2 Report

The paper is an interesting and comprehensive review of vulvovaginal candidiasis (VVC) caused by Candida albicans in light of human vaginal microbiota and host immunity. The topic is very complex, but the authors managed to present it intelligibly. The authors discussed several treatment options in response to the last research advances in the pathogenesis of VVC. 

Minor comments:

Line 313: for a wide audience, please, explain what is anakinra (IL-1Ra).

Figure 1. Please, explain the abbreviation SCFA in the figure caption.

Line 346: pH, instead of PH.

Lines 356, 437: the impact of H2O2 produced by vaginal lactobacilli is under debate https://pubmed.ncbi.nlm.nih.gov/29409534/. Please, discuss in the paper.

Line 632: for a wide audience, please, explain what is leukotriene.

Author Response

Comments from reviewer 2

R.: The paper is an interesting and comprehensive review of vulvovaginal candidiasis (VVC) caused by Candida albicans in light of human vaginal microbiota and host immunity. The topic is very complex, but the authors managed to present it intelligibly. The authors discussed several treatment options in response to the last research advances in the pathogenesis of VVC.

A.: Dear reviewer, thank you very much for your review and valuable comments. As you requested, we have made all necessary changes in our manuscript. The changes are highlighted in blue.

Minor comments:

R.: Line 313: for a wide audience, please, explain what is anakinra (IL-1Ra).

  1. Anakinra is specified as follows “anakinra, a recombinant interleukin-1 receptor antagonist (IL-1Ra)”. Please see paragraph 3.2, line 323-324.

R.: Figure 1. Please, explain the abbreviation SCFA in the figure caption.

A.: In the figure 1 caption “SCFA” is specified as follows: “short chain fatty acids”

R.:  Line 346: pH, instead of PH.

A.: We apologize for this error. We have corrected “pH”. Please see line 357 

R.: 356, 437: the impact of H2O2 produced by vaginal lactobacilli is under debate https://pubmed.ncbi.nlm.nih.gov/29409534/. Please, discuss in the paper.

A.: Thank you for this comment. We have discussed the impact of H2O2 on C. albicans by referring to the article “https://pubmed.ncbi.nlm.nih.gov/29409534”,  as suggested. Please see paragraph 4.1, lines 367-375. We have deleted the following sentence ”Moreover, H2O2 and biosurfactants produced by lactobacilli display a strong antifungal effect [111]. In particular, H2O2 was found to inhibit Candida overgrowth as well as its yeast-to-hyphal dimorphic transition [110]”

R.: Line 632: for a wide audience, please, explain what is leukotriene.

A.: “leukotrienes” are specified as follows “a group of inflammatory vasoactive mediators”. Please see paragraph 6, line 717.

Author Response

Comments from reviewer 3

Dear Editor, Thank you for your peer review invitation. The manuscript, entitled “The interplay between Candida albicans, host immunity and vaginal microbiota in health and disease: an overview and future perspectives” reviewed the latest advances in the pathogenic mechanisms implicated in the onset of VVC and also discusses novel potential strategies, with a special focus on the use of probiotics and vaginal microbiota transplantation in the treatment and/or prevention of recurrent VVC. The topic of this review is interesting, however, due to some drawbacks, my suggestion is major revision.

R.:  1. I suggest the authors to pay attention on the Candida albicans strains/isolates used in the reference literatures included in this review. There would be huge difference in the findings if standard strain or clinical isolates were used. I suggest the authors to pay more attention on the difference and focus on the studies using clinical isolates, which reflect the real situations. For example, the toxin candidalysin from different isolates could be totally different (https://doi.org/10.1371/journal.ppat.1009884), due to the difference in ECE1 gene

A.: Dear reviewer, we wish to express our appreciation for your in-depth comments and suggestions which have greatly improved the manuscript. We have made changes according to your suggestions. All the changes are highlighted in red.

We have focused our attention, in different parts of the manuscript, on the studies performed by using clinical isolates of C. albicans. Please see lines:

2.2 lines 156-167

4.2 lines 497-501 (already discussed in the original version of the manuscript)

5.2. lines 576-578

5.3 lines 652-657

5.4 lines 681-690

R.:2. Concerning the interplay between Candida albicans and vaginal microbiota, the vaginal microbial communities in VVC patient and their comparison with those in healthy women are also important.

A.: Regarding the vaginal microbial communities in VVC patients and their comparison with those in healthy women, we have addressed this point in paragraph 4.2, lines 443-446. This issue has already been discussed in the original version of the manuscript. Please see paragraph 4.2 lines 449-452 and lines 474-481.

R.: 3. The part 5 C. albicans-related key virulence factors in the pathogenesis of VVC/RVVC is too short and does not contribute much to the review.

A.: We have expanded and reorganized this section by including the new advances on the mechanisms of action of C. albicans candidalysin and the impact of candidalysin gene polymorphisms on Candida pathogenicity during VVC. Please see paragraph 5.2. lines 552-580.

The following part has been removed “Candidalysin is a cytolysin belonging to candidalysins, a new family of fungal peptide toxins [37] that is encoded by the ECE1 gene. Candidalysin was the first cytolytic peptide toxin to be identified in human fungal pathogens. It is produced by targeted proteolytic processing of the hypha-associated protein Ece1p, and following secretion, it plays a critical role in VVC immunopathology. Its key role has been clearly demonstrated in mice unable to express and secrete this cytolysin that were challenged with fungal strains, as a significant reduction in VEC damage, neutrophil recruitment and pro-inflammatory cytokine production were observed in the mice [36,162]. This peptide not only induces direct VEC damage but also triggers innate immune responses in epithelial cells, mainly through MAPK signalling, which activates the p38 and ERK1/2 pathways, and these, in turn, activate the AP-1 transcription factors c-Fos and MAPK phosphatase 1, respectively [37,163], leading to the secretion of an array of antimicrobial peptides, alarmins and pro-inflammatory cytokines, with consequent recruitment of immune cells to the infection foci”

We have also added a new paragraph on the potential role of beta-glucan unmasking in the pathogenesis of VVC (please, see 5.4, lines 660-694).

R.: 4. The conclusion is too lengthy. I suggest the authors to shorten it to one paragraph and only include the major conclusions and the author’s opinion on future perspectives.

A.: As suggested, we have rewritten the conclusions and perspectives in a separate paragraph (please see the paragraph 7).

R.: 5. Please go through the whole text and double check on the formatting. For example, species names should be italic.

A.: We have carefully checked the whole text and corrected the errors, including the species names.

Round 2

Reviewer 3 Report

The manuscript, named The interplay between Candida albicans, host immunity and vaginal microbiota in health and disease: An overview and future perspectives has been revised by the authors carefully. Due to some drawbacks, my suggestion is major revision.

Comments:

1. As the section 2 gave some information about the effect of VEC on VVC, the title of this review “The interplay between Candida albicans, host immunity and vaginal microbiota in health and disease: An overview and future perspectives” does not summarize the whole text comprehensively.

2. It is suggested to use more figures and tables to show the information to make the information more visual. For example, for the information in section 2. Role of vaginal epithelial cells in the pathogenesis of VVC , the authors could use a figure to conclude the relevant information, same to section 3.

3. The legend of figure 1 does not explain the content of this figure. It is suggested to introduce the three parts in detail.

4. In section of 5.3 biofilm formation, in vaginal C. albicans normally form biofilm with other bacteria, so, it is suggested to supplement relevant information on two or more bacteria in biofilm.

5. It is suggested to summarize the information of 6. Potential therapeutic strategies for the preven- 696 tion and/or treatment of VVC in a table.

6. This review is about Candida albicans, but in line 30, it says “but many other non-albicans Candida species, including C. glabrata, C. krusei, C. parapsilosis and the most recently discovered multi-drug resistant C. auris strain, may also cause human infections”. It is suggested to delete this sentence, as it makes the readers think that the following content is about non-albicans.

Author Response

The manuscript, named “The interplay between Candida albicans, host immunity and vaginal microbiota in health and disease: An overview and future perspectives” has been revised by the authors carefully. Due to some drawbacks, my suggestion is major revision.

A.: Dear Reviewer, we wish to express our appreciation for your in-depth comments and suggestions which have greatly improved the manuscript. We have made changes according to your suggestions. All the changes are highlighted in red.

Regarding your comment on quality of English we would like to clarify that the manuscript has been carefully revised by a professional language editing service before the first submission, and by a native-English speaker, qualified to write medical and scientific manuscripts, during the first round of revision.

Comments:

  1. : As the section 2 gave some information about the effect of VEC on VVC, the title of this review “The interplay between Candida albicans, host immunity and vaginal microbiota in health and disease: An overview and future perspectives” does not summarize the whole text comprehensively.

A.: We agree with Reviewer 3. The title “The interplay between Candida albicans, host immunity and vaginal microbiota in health and disease: An overview and future perspectives” has been replaced with “The interplay between Candida albicans, vaginal mucosa, host immunity and resident microbiota in health and disease: An overview and future perspectives”.

  1. : It is suggested to use more figures and tables to show the information to make the information more visual. For example, for the information in section “2. Role of vaginal epithelial cells in the pathogenesis of VVC ”, the authors could use a figure to conclude the relevant information, same to section 3.

A.: Thank you for your suggestion. We have added two new Figures: Figure 1 that is a schematic representation of the principal effects resulting from the C. albicans hyphae-VECs interaction in VVC, and Figure 2 that summarizes the role of neutrophils and macrophages in the immunophatogenesis of VVC.

  1. : The legend of figure 1 does not explain the content of this figure. It is suggested to introduce the three parts in detail.

A: We apologize for being unclear. The legend of Figure 3 (ex Figure 1) has been rewritten more clearly.

  1. : In section of “5.3 biofilm formation”, in vaginal C. albicans normally form biofilm with other bacteria, so, it is suggested to supplement relevant information on two or more bacteria in biofilm.

A.: This issue has been addressed in the paragraph 5.3.  Please see lines 694-709.

  1. : It is suggested to summarize the information of “6. Potential therapeutic strategies for the preven- 696 tion and/or treatment of VVC” in a table.

A.: Thank you for your suggestion. The potential strategies for the prevention/treatment of VVC/RVVC  have been summarized in Table 1.

  1. : This review is about Candida albicans, but in line 30, it says “but many other non-albicans Candida species, including C. glabrata, C. krusei, C. parapsilosis and the most recently discovered multi-drug resistant C. auris strain, may also cause human infections”. It is suggested to delete this sentence, as it makes the readers think that the following content is about non-albicans.

A: The sentence “but many other non-albicans Candida species, including C. glabrata, C. krusei, C. parapsilosis and the most recently discovered multi-drug resistant C. auris strain, may also cause human infections” has been deleted as suggested. Please see line 30.
